# Bacterial Adhesion on Dental Polymers as a Function of Manufacturing Techniques

**DOI:** 10.3390/ma16062373

**Published:** 2023-03-16

**Authors:** Jörg Bächle, Cordula Merle, Sebastian Hahnel, Martin Rosentritt

**Affiliations:** Department of Prosthetic Dentistry, UKR University Hospital Regensburg, 93043 Regensburg, Germany

**Keywords:** bacterial adhesion, CAD/CAM, 3D printing, PAEK, PEEK, PEKK, AKP, composites, PMMA, roughness, surface free energy

## Abstract

The microbiological behavior of dental polymer materials is crucial to secure the clinical success of dental restorations. Here, the manufacturing process and the machining can play a decisive role. This study investigated the bacterial adhesion on dental polymers as a function of manufacturing techniques (additive/subtractive) and different polishing protocols. Specimens were made from polyaryletherketone (PEEK, PEKK, and AKP), resin-based CAD/CAM materials (composite and PMMA), and printed methacrylate (MA)-based materials. Surface roughness (R_z_; R_a_) was determined using a laser scanning microscope, and SFE/contact angles were measured using the sessile drop method. After salivary pellicle formation, in vitro biofilm formation was initiated by exposing the specimens to suspensions of *Streptococcus mutans* (*S. mutans*) and *Streptococcus sanguinis* (*S. sanguinis*). Adherent bacteria were quantified using a fluorometric assay. One-way ANOVA analysis found significant influences (*p* < 0.001) for the individual parameters (treatment and material) and their combinations for both types of bacteria. Stronger polishing led to significantly (*p* < 0.001) less adhesion of *S. sanguinis* (Pearson correlation PC = −0.240) and *S. mutans* (PC = −0.206). A highly significant (*p* = 0.010, PC = 0.135) correlation between *S. sanguinis* adhesion and R_z_ was identified. Post hoc analysis revealed significant higher bacterial adhesion for vertically printed MA specimens compared to horizontally printed specimens. Furthermore, significant higher adhesion of *S. sanguinis* on pressed PEEK was revealed comparing to the other manufacturing methods (milling, injection molding, and 3D printing). The milled PAEK samples showed similar bacterial adhesion. In general, the resin-based materials, composites, and PAEKs showed different bacterial adhesion. Fabrication methods were shown to play a critical role; the pressed PEEK showed the highest initial accumulations. Horizontal DLP fabrication reduced bacterial adhesion. Roughness < 10 µm or polishing appear to be essential for reducing bacterial adhesion.

## 1. Introduction

Composites, PMMA, and thermoplastic polymer materials such as polyaryletherketones (PAEKs) represent the state of the art for the fabrication of temporary and fixed dental prostheses [1,2]. With regard to manufacturing processes in dental technology, additive (AM) and subtractive manufacturing (SM) techniques are increasingly replacing conventional manufacturing protocols. It has been shown that the manufacturing methods have an influence on the mechanical properties [3,4]. Thus, it can be assumed that the fabrication may also influence bacterial adhesion.

Oral biofilm formation follows a sequential colonization that begins with the adhesion of early colonizers such as streptococci to the pellicle-coated surface and is strongly influenced by material-dependent surface properties [5,6]. Plaque and biofilm formation can lead to caries formation and periodontal inflammation [7,8]. Secondary caries is still considered the main reason of dental restauration failure [9]. Relevant parameter for the formation of dental biofilms on dental materials include the type of material and its chemical composition [10,11]. Surface properties such as roughness, topography, and surface free energy also play a major role in the adhesion of dental biofilm. The conventional wisdom is that low surface roughness and high surface free energy minimize biofilm formation, although—in dentistry and for dental materials—ideal surfaces in complex materials have not yet been identified [12]. Nevertheless, a profilometrically determined roughness value of 0.2 µm is commonly considered as a threshold to limit oral biofilm formation [13].

SM resin-based systems are produced under industrial conditions and guarantee improved mechanical properties [2,14]. Milled CAD/CAM materials provide a varying modulus of elasticity ranging between 2–3 GPa (PMMA) and 10–18 GPa (highly filled resin-based composite). It has been highlighted that the adhesion of different bacterial strains is strongly influenced by the individual composition [15,16,17].

AM of methacrylate (MA) materials represents an interesting and cost-saving manufacturing alternative. However, lower mechanical strength of the processed products must be expected, which may also depend on manufacturing parameters such as layer thickness or postprocessing [18,19]. An influence of the printing direction on the mechanical properties has been identified [20,21,22,23]. Furthermore, building angle is supposed to impact bacterial adhesion [24]. Surface roughness and surface free energy have been demonstrated to influence microbial adhesion [15,25]. SM polymers are expected to show reduced bacterial adhesion [26].

On the basis of their chemical composition, PAEKs can be divided into three subgroups: PEEK (polyetheretherketone), PEKK (polyetherketoneketone), and AKP (arylketonepolymer). Their mechanical properties (E-modulus of 3.5–5.1 GPa) are influenced by the relative proportion of ketone and ether groups [27]. Titanium dioxide supplementation up to 10–30 wt.% enhances strength, modulus, and stiffness. PAEKs may be processed with different manufacturing techniques including milling, vacuum pressing from pellets or granules, and fused filament fabrication [28]. PAEKs contain no MA monomers, are free of metals, and are chemically inert, which makes them interesting alternatives for patients with allergies [29] and for a variety of clinical applications in prosthetic dentistry such as fixed and removable dental prostheses or implant abutments [29,30,31]. Regarding bacterial adhesion and biofilm formation on PAEKs, scientific evidence is limited [1,32].

Against this background, the aim of this study was to examine the effects of surface free energy and surface roughness of different CAD/CAM-fabricated polymer materials on the bacterial adhesion of *S. mutans* and *S. sanguinis*. The null hypotheses were that bacterial adhesion is not influenced by the type of material, the individual fabrication process, or surface free energy and surface roughness.

## 2. Materials and Methods

Specimens (*n* = 90 per material, 8 mm diameter, 2 mm height) were fabricated from 12 dental restorative materials. Specimens were fabricated from polyaryletherketone (PAEK), various CAD/CAM materials (composite and temporary restorative materials), polymethylmethacrylate (PMMA), and an AM MA material to be used in fixed dental prosthesis fabrication. PAEK materials differed in their composition and fabrication mode (milling, press technique, injection molding, and fused filament fabrication). AM specimens were printed in different directions to the building platform (0°/90°) to investigate the influence of fabrication (Table 1).

PMMA and composite CAD/CAM specimens were milled (inLAB MC X5, Sirona Dentsply, Bensheim, Germany). PEAK specimens were milled (M2, Zirkonzahn, Gais, Italy), pressed (Bredent for 2 Press, Bredent, Senden, Germany), or printed (fused filament fabrication, Vestakeep i4 3DF, Apium P220, Apium Additive Technologies, Karlspark, Germany). The AM specimens were printed in vertical or horizontal direction (DLP printer Formlabs form 3B, Permanent Crown, Formlabs, Somerville, MA, USA, layer 50 µm). The study design is shown in Figure 1. For all materials, three surface protocols (*n* = 30) were applied:P0: no polishing; surface properties as produced by the fabrication process itself;P1000: polishing (Tegramin-25, Struers, Willich, Germany, 100 rpm water cooling) for 30 s (silicon-carbide grinding paper 1000 grid, Stures, Willich, Germany);P4000: polish protocol P1000 + polishing (Tegramin-25, Struers, Willich, Germany, 100 rpm water cooling) for 30 s (silicon-carbide grinding paper 4000 grid, Buehler, Düsseldorf, Germany).

Glass and a veneering composite (Sinfony, 3M, Seefeld, Germany; polished for 2 × 30 s, P1000/P4000, Buehler, Düsseldorf, Germany) were used as a reference.

**Surface characterization:** Roughness (R_z_, R_a_) on all specimens was measured with a laser scanning microscope (KJ VK-X100K, Keyence, J, class 2 laser 658 nm, 50× magnification; measuring range 2500 × 1900 µm, cutoff λs 0.8 µm λc 0.08 µm, resolution h = 0.005 µm, w = 0.01 µm, 50 × 12 nm, repeatability σ h: 0.02 µm, w: 0.05 µm). Contact angles with two liquids differing in hydrophobicity (Millipore water, Diiodomethane) were determined using the sessile drop method and a computer-aided contact angle measurement device (DSA25, Krüss, Hamburg, Germany, drop volume of 1 µL). Surface free energy (SFE) was calculated. SEM micrographs (Phenom, FEI Company, Eindhoven, The Netherlands) were used to assess the surfaces prior and after polishing (untreated specimens, resolution 1024, quality high, WD 10 µm, magnifications up to 1500×, *n* = 2).

**Biofilm formation:** A fluorometric assay (Resazurin reduction, Alamar Blue) was used to investigate the adhesion of two Gram-positive oral bacteria *(S. mutans* and *S. sanguinis)* [33,34,35].

The blue and nonfluorescent Resazurin is reduced by viable and metabolically active cells/bacteria via dehydrogenase enzymes into the violet and fluorescent pigment Resorufin. Quantification was performed by measuring fluorescence or absorbance, which is linearly proportional to the number of living bacteria in the sample.

All specimens were cleaned with 70% *v*/*v* ethanol for 10 s and then rinsed with distilled water for 10 s. Specimens were transferred into sterile 48-well plates, fixed, and incubated with 1 mL of phosphate-buffered saline solution (PBS, Sigma-Aldrich, St. Louis, MO, USA). The intrinsic fluorescence of the specimens was determined.

Pellicle formation was initiated by incubating the sample-containing plates with 1 mL of artificial saliva per well at 37 °C in a thermos-shaking-device (OrbitalShaker; Thermo Forma, Marietta, OH, USA). After 2 h, saliva was carefully removed, and each specimen was incubated with 1 mL of microbial suspension (either *S. mutans* or *S. sanguinis*) and 15 µL of Resazurin (Resazurin, Sigma-Aldrich, St. Louis, MO, USA) for 2.5 h. Specimens were gently rinsed twice with 1 mL of PBS to remove unbound bacteria. Fluorescence was measured using an automated multi-detection reader (Fluostar Optima, BMG Labtech, Offenburg, Germany).

Statistical analysis was performed with SPSS 28.0 (IBM, Armonk, NY, USA). Normal distribution of data was verified using the Shapiro–Wilk test. Means and standard deviations were calculated and analyzed using one-way analysis of variance and the Bonferroni test for post hoc analysis. Between-subject effects were investigated. The level of significance was set to α = 0.05. Pearson correlation (PC) between the individual parameters was determined.

## 3. Results

### 3.1. Bacterial Adhesion

The Shapiro–Wilk test confirmed a normal distribution of the data in ~80% of cases. One-way ANOVA confirmed significant influences (*p* < 0.001) for individual parameters treatment and material, as well as their combinations. The highest impact on the adhesion of *S. mutans* was found for the combination of material and treatment (η^2^ = 0.879, *p* < 0.001), followed by material (η^2^ = 0.860, *p* < 0.001) and treatment (η^2^ = 0.589, *p* < 0.001). For the adhesion of *S. sanguinis*, material (η^2^ = 0.600, *p* < 0.001) had the highest impact, followed by the combination of material and treatment (η^2^ = 0.595, *p* < 0.001), while treatment had a significantly lowest impact (η^2^ = 0.317, *p* < 0.001).

#### 3.1.1. *S. mutans*

Mean relative fluorescence intensities varied between 36 for TC and 2043 for MA-V. Post hoc analysis identified that PEKK and TC showed significantly (*p* < 0.037) lower fluorescence intensities correlating with less bacterial adhesion compared to printed specimens (MA-V, MA-H, and A) and pressed polyetheretherketone (P, P plus), as well as IM-E and the veneering composite (Table 2).

The lowest fluorescence intensities indicating lowest microbial adhesion were identified for PEKK for all unpolished materials, while TC showed the lowest fluorescence intensities correlating with lowest bacterial adhesion for polishing protocols. However, no significant (*p* = 1.000) difference between milled specimens was observed. MA printed specimens (MA-V, MA-H) featured significantly (*p* < 0.001) higher fluorescence intensities indicating higher bacterial adhesion in comparison to the other materials.

#### 3.1.2. *S. sanguinis*

Mean relative fluorescence intensities ranged between 83 for AKP and 1216 for MA-V. Post hoc analysis revealed significantly (*p* < 0.001) lower fluorescence intensities indicating lower biofilm adhesion for AKP compared to BC, as well as both pressed PEEK (P, P plus) and MA printed specimens (MA-V, MA-H). Pressed PEEK specimens showed significantly (*p* < 0.001) higher fluorescence intensities correlating with higher bacterial adhesion compared to both control groups (glass and Sinfony) and milled specimens (TC, AKP, PEKK, and PEEK) as well as injection-molded and printed PEEK specimens. AKP featured lowest relative fluorescence intensities indicating lowest biofilm formation for all polishing protocols. MA-V showed significantly (*p* < 0.001) higher fluorescence intensities indicating higher adhesion of bacteria compared to all other materials, while no significant (*p* = 1.000) differences were observed between MA-H and pressed PEEK (P and P plus) specimens Table 2.

### 3.2. Influence of Fabrication Methods

Post hoc analysis revealed that milled PEEK specimens showed no significant (*p* = 1.000) difference in fluorescence intensities indicating the adhesion of *S. sanguinis* when compared to injection molded specimens (IM-E, IM) and printed PEEK (A). Pressed specimens (P, P plus) showed significant (*p* < 0.001) more *S. sanguinis* adhesion than the other manufactured PEEK specimens. Post hoc analysis of *S. mutans* adhesion revealed no significant (*p* = 0.145) differences for different fabrication methods of PEEK. Printed MA specimens which were printed in horizontal direction (MA-H) promoted significantly (*p* < 0.001) less biofilm compared to specimens with vertical printing orientation (MA-V).

### 3.3. Effects of Polishing

Post hoc analysis with homogeneous subsets revealed significant (*p* < 0.001) differences between the polishing procedures P0 and P1000, as well as between P0 and P4000, regarding relative fluorescence intensities, indicating differences in microbial adhesion.

### 3.4. Surface Parameters—Roughness and SFE

The results of the surface roughness measurements R_z_ and R_a_, as well as SFE, measured for each material and polishing protocol, are displayed in Table 3.

P0: The highest mean R_z_ values were measured for PEKK (R_z_ = 134 ± 7.9 µm; R_a_ = 13.8 ± 0.5 µm). A showed the lowest roughness values (R_z_ = 59.7 ± 12.2 µm; R_a_ = 3.6 ± 0.9 µm). Mean roughness values were R_z_ = 81 ± 22.2 µm and R_a_ = 5.5 ± 2.7 µm. Surface free energy ranged between 32.9 mJ/m^2^ (PEKK) and 58.55 mJ/m^2^ (A).

P1000: After polishing with grid 1000, surface roughness varied as follows: R_z_ = 52.1 ± 4.4 (BC) to 84 ± 7.7 µm (PEEK) and R_a_ = 3.8 ± 0.3 (BC) to 9 ± 1.9 µm (PEEK). Mean roughness values were R_z_ = 65.5 ± 9.8 µm and R_a_ = 5.9 ± 1.9 µm. SFE varied from 36.84 (BC) to 48.36 mJ/m^2^ (A).

P4000: The highest surface roughness was identified for A (R_z_ = 58.6 ± 8.0 µm; R_a_ = 6.7 ± 0.9 µm), while P showed the lowest surface roughness (R_z_ = 27.7 ± 3.3 µm; R_a_ = 6.7 ± 0.9 µm). Mean roughness values were R_z_ = 40.4 ± 9.6 µm and R_a_ = 4.5 ± 1.2 µm. SFE ranged between 32.39 (BC) and 49.47 mJ/m^2^ (TC).

ANOVA analysis showed significant (*p* < 0.002) differences in relative fluorescence intensities indicating differences in bacterial adhesion as well as significant (*p* < 0.001) differences in surface roughness (R_z_ and R_a_).

### 3.5. SEM Figures

In the initial condition P0, the specimens exhibited a wavy and slightly rough, but mostly homogeneous surface. IM-E showed isolated cracks, and P Plus exhibited a heavily damaged surface. After the first machining P1000, the specimens mostly showed grinding marks and occasional scratches and cracks. IM-E had isolated holes. P Plus showed a homogeneous surface with punctual brightening. After the second processing step P4000, mostly homogeneous surfaces with isolated scratches were found. P Plus and MA-H also showed additional grinding marks (Figure 2).

### 3.6. Correlations

A highly significant (*p* = 0.010, PC = 0.135) correlation between relative fluorescence intensities and R_z_ was identified in the *S. sanguinis* model, while no significant (*p* = 0.138) correlations between R_z_ and relative fluorescence intensities was identified in the *S. mutans* model. Regardless of the individually applied microbial model, no significant correlations between R_a_ (*p* = 0.273) and SFE (*p* = *0*.0667) and fluorescence intensities were identified.

A significant (*p* < 0.001, PC = 0.819) correlation between R_z_ and R_a_ was shown. Polishing provided a significant (*p* < 0.001) negative correlation for surface parameters R_z_ (PC = −0.739) and R_a_ (PC = −0.439). No significant correlation between SFE (*p* = 0.379) and surface treatment was identified. Prolonged polishing protocols significantly (*p* < 0.001) reduced biofilm formation of *S. sanguinis* (PC = −0.240) and *S. mutans* (PC = −0.206).

## 4. Discussion

### 4.1. Type of Material

The first null hypothesis suggesting that the type of material has no influence on bacterial adhesion has to be rejected. The investigated materials showed different adhesion of streptococci. For about 50% of the materials, microbial adhesion was considerably lower or about twice as high as for the reference materials. Exceptions were unpolished AM MA samples which exceeded microbial adhesion to the reference materials manyfold.

PAEKs: For the various PAEK materials, significant differences in microbial adhesion were identified. Similar results were previously reported for other thermoplastic materials [36,37]. Despite the same manufacturing method, the SM-processed PAEK materials showed different bacterial adhesion. These differences were probably due to the composition of the various PAEK materials. PAEK systems vary in their degree of crystallinity and their rate of crystallization, both depending on the individual technique blanks and semi-finished products. PEEK is generally amorphous or semi-crystalline, but it can also reach a very high degree of crystallinity (up to 40%). Processing temperatures during pressing or SM may impair the surface quality. Surface defects on SM PEEK could only be removed after polishing with 4000 grit sandpaper. In contrast to composites, the type and amount of filler in thermoplastic materials had only a small influence on bacterial adhesion, which might be attributed to the small quantities involved. Nevertheless, microbial adhesion on PEKK might have been influenced by nano spikes, as reported in previous studies [38,39].

Composite: The resin-based CAD materials and the laboratory reference composite showed a low tendency to bacterial adhesion. These observations can be explained by the composition of the filled resin-based materials. Minimizing resin matrix exposure might reduce biofilm formation on the surface of resin composites [17]. A negative correlation between inorganic filler content and biofilm formation was previously identified [10]. The highly filled composite showed higher bacterial adhesion than the unfilled PMMA, but in the range of the less highly filled printed materials. Moktar et al. revealed that a nanohybrid filler system and a modified resin matrix structure cause considerably less biofilm formation [15]. Further research should, therefore, address the influence of filler content on bacterial adhesion even in thermoplastic formulations.

AM materials: The printed samples showed higher *S. mutans* accumulation compared to the controls. For printability, smaller and fewer fillers, as well as a modified matrix, are required, which presumably impacts the adhesion of microorganisms. Unpolished AM MA specimens showed significantly higher adhesion of microorganisms compared to polished specimens, which is most likely caused by uncured resin remnants on the surface of the specimens. Post-polymerization or washing procedures have been shown to influence the in vitro cytotoxicity and, therefore, might also affect bacterial adhesion [40,41]. This study also observed a tendency for increased bacterial adhesion on printed MA samples that were subjected to the polishing protocol P4000.

### 4.2. Fabrication of the Material

The null hypothesis that fabrication methods do not influence bacterial adhesion has to be partly rejected. The data of the current study indicated that the effect of fabrication methods is strain-dependent, as *S. mutans* adhesion was not affected by fabrication methods. For *S. sanguinis*, significantly higher adhesion was identified on pressed PEEK specimens than on specimens processed by the other fabrication methods (milling, injection molding, and printing). As a result of polishing, processing effects were significantly altered. Relevant differences in microbial adhesion between milled, injection-molded, and pressed PEEK were identified. Artefacts such as air inclusions could be observed for pressed and injection-molded specimens. These superficial defects might explain the higher bacterial adhesion (especially for pressed specimens). Industrial fabrication of blanks increases mechanical properties [3] and, probably, the surface quality. It is suggested that there is a reduced risk of porosities when the material is fabricated under optimal conditions [20]. A comparable effect was also observed for industrially fabricated resin composite blocks for dental CAD/CAM applications [42]. All materials processed with CAM featured similar levels of bacterial adhesion compared to the other manufacturing processes. Nevertheless, previous studies did not report a noticeable difference in the adhesion of *Staphylococcus aureus* and *Candida albicans* to pressed or milled PEEK samples [26].

Temperature management and cooling processes are crucial for the structure and quality of semi-crystalline materials, as they influence the composition of the crystalline and amorphous phases. The processes in which the material is formed during heating (fused filament fabrication and pressing) are likely to be more affected. When shaping was carried out in the CAM process, a low level of bacterial adhesion was observed. This phenomenon could have been caused by modification of the surface due to crystallization effects. Thus, the temperature control during processing and, for example, the preheating of the muffle could also have an effect on the structure of the thermoplastic material. Since the temperature control in the blank plays an important role, the shape and size of the blank are also relevant. In addition, the type of mold used (plaster or metal) could lead to interactions and/or changes in the surface, thus affecting the adhesion of *S. sanguinis*. Higher adhesion of *S. sanguinis* was identified in specimens produced by the compression molding process than in those processed using injection molding. During extrusion, the solidification of the filament results in a partial alignment of the polymer chains, whereby, starting from the crystallization nuclei, molecule chains become structured and form ordered regions. The deposition in filament production is in the middle range between CAD and press production. The influence of the adhesion mechanisms of the different bacterial strains was evident in the adhesion of *S. mutans*.

### 4.3. AM Systems

Comparing different building angles, horizontally printed MA specimens promoted less biofilm formation compared with vertically printed specimens. The reason for this could be that, in vertical production, a number of layers and transitions are on the measuring surface of the specimens, whereas, in horizontal production, only one layer is exposed on the surface. Mokhtar et al. observed that bacterial adhesion on MA materials was influenced by fabrication methods (AM, SM, and heat/cold curing), with AM showing the best results [15]. Results were heterogenous, since more C. albicans was observed on AM specimens compared to SM ones [43]. Although a significant effect of the building angle on surface roughness/topography was previously reported, no relevant difference in the adhesion of *Candida albicans* was observed [24].

### 4.4. Surface Free Energy and Roughness

The influence of the superficial roughness on the adhesion of microorganisms has been intensively discussed in the literature [25,26,44,45], and it always represents the requirement for a smooth surface [46]. It is expected that microbial adhesion can modify fillers and reduce the microhardness of the surface. The release of material components, especially ethylene glycol dimethacrylate or triethylene glycol dimethacrylate, can also promote the growth of further bacteria [47,48].

R_a_: The null hypothesis that bacterial adhesion is not influenced by surface roughness R_a_ could be confirmed. R_a_ showed no correlation with bacterial adhesion, although the materials provided significantly different initial values in R_a_ in the unpolished state and after both polishing steps. One reason for this is that, especially in the low roughness range of R_a_, the application of the pellicle caused a leveling and equalization of the surfaces. For most materials, lowest R_a_ roughness was achieved with the P4000 polishing protocol. Exceptions were AKP and IM-E, which hardly showed any R_a_ changes after the various polishing regimes, and A, which was even rougher after polishing. These observations can be explained by the distinct type of fabrication, as the materials showed smooth surfaces directly after production. For material A, this was certainly due to the filament printing process, where the processing parameters, such as temperature and the type of die through which the molten material is pressed, determine the surface quality. For the AKP specimens, a different polymer matrix compared to other PAEKs was expected. For IM-E, the relatively high filler content for PEEK influenced the surface finish.

The previously introduced R_a_ roughness threshold of 0.2 µm was highly exceeded by all materials of the current study, due to the acquisition of surface roughness data. In comparison to mechanical scanning, the surface quality, such as reflecting, light-absorbing, or absorbing surfaces, can result in significant differences in the roughness of various materials. For surface roughness measurements, discrepancies of about 75% of the value obtained between the stylus method and other systems have been reported [49,50,51]. The samples all fell within the range of the clinically relevant reference veneering material. These considerations confirm that the threshold value is dependent on the measurement approach applied.

R_z_: The hypothesis that bacterial adhesion is not affected by surface roughness R_z_ could only be partially confirmed. A positive correlation between R_z_ and microbial adhesion was shown for *S. sanguinis*. This observation might partly confirm earlier studies, which reported a positive correlation between surface roughness and bacterial adhesion [10,12,17,25,26]. In most cases, polishing resulted in a decrease in R_z_ values, with the lowest values identified after a 4000-grit polishing regime. Overall, this polishing regime caused significantly less adhesion of *S. sanguinis* and *S. mutans*.

SFE: The null hypothesis that SFE did not influence bacterial adhesion could be confirmed. SFE was not correlated with either polishing or bacterial adhesion. In contrast to previous studies, it could not be shown that the surface energy of the materials affects the adhesion of the bacteria [52]. SFE was not affected by processing, as shown for MA or PEAK materials, but might have been influenced by filler or matrix components. Protein formation and bacterial adhesion decrease in materials with a high surface energy [53,54]. It is well known that the formation of pellicles reduces the surface energy on the tooth surface [52,53]. Due to the formation of pellicles, the free surface energies of the different surfaces approach each other [53]. The correlation between SFE and bacterial adhesion appears to be strain-specific; hydrophilic bacteria such as *S. mutans* with a high surface energy tend to be hydrophilic surfaces [55,56,57].

## 5. Conclusions

Resin-based materials, composites, and PAEKs showed noticeable differences in bacterial adhesion. Fabrication methods were shown to play a critical role for bacterial adhesion; the pressed PEEK exhibited the highest initial accumulation, whereas the milled PAEK samples provided similar lower adhesion. Horizontal DLP fabrication reduced bacterial adhesion in comparison to vertical processing. For DLP printed MA and pressed PEEK, the surface finishing significantly reduced bacterial adhesion, making it essential for clinical application. The processing method and the polish affected the quality of the surfaces; Apium and Ultraire already have very smooth surfaces due to processing and, thus, low bacterial adhesion. Roughness values < 10 µm or polishing appear to be essential for reducing bacterial adhesion, whereas SFE did not show any influence.

## Figures and Tables

**Figure 1 materials-16-02373-f001:**
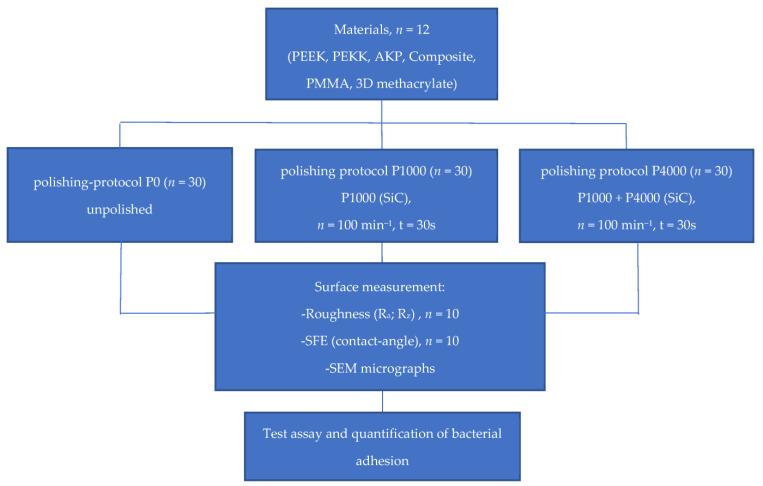
Study design.

**Figure 2 materials-16-02373-f002:**
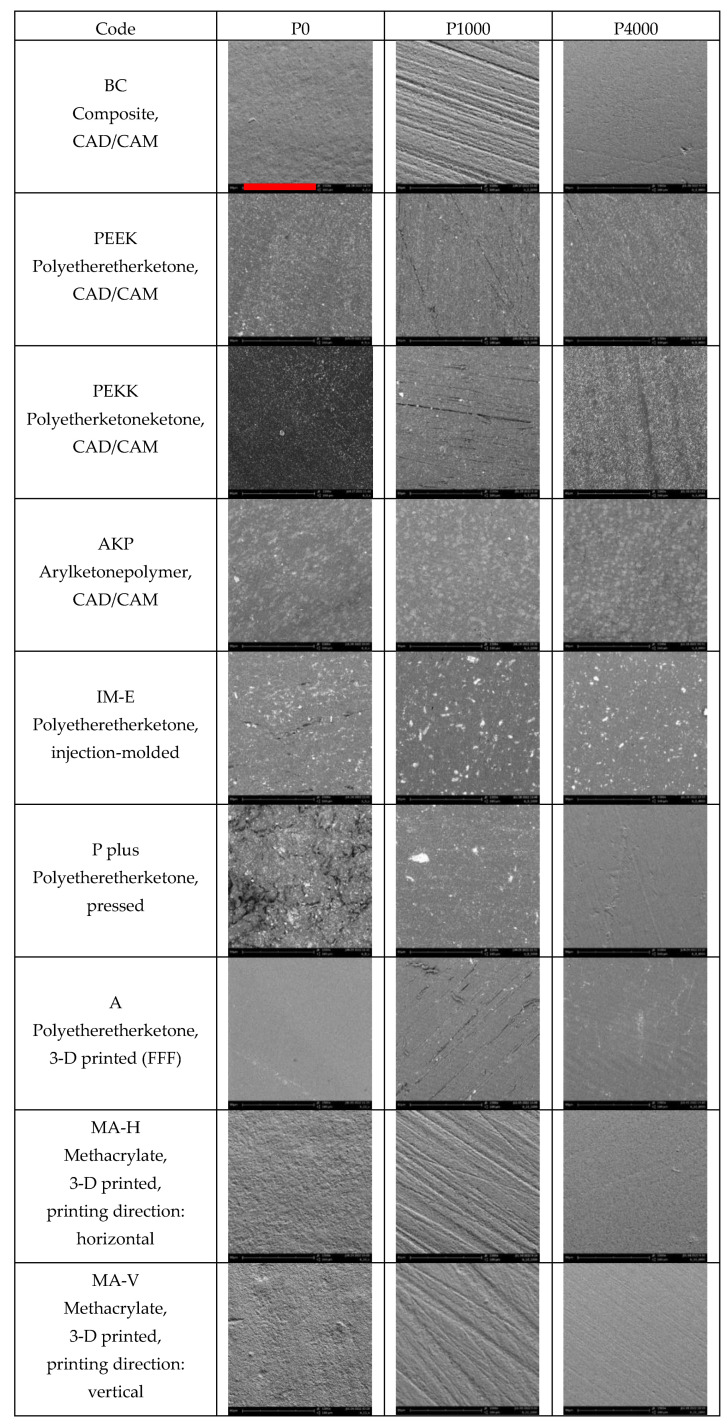
SEM-images, magnification 1500× (red bar: 80 µm valid for every figure).

**Table 1 materials-16-02373-t001:** Name, abbreviation, manufacturer, type, composition, mechanical properties, and fabrication method of materials (-: no values available, PEEK: polyetheretherketone).

Name	Abbr.	Manufacturer	Type	Filler (wt.%)	E (GPa)	FS (MPa)	Fabrication Method
Telio CAD	TC	Ivoclar Vivadent, Schaan, Liechtenstein	Polymethylenemethacrylate (PMMA)	No filler	2.8	135	Milling, CAD/CAM
Brilliant Crios	BC	Coltene, Altstätten, Switzerland	Composite	72%	10.3	198	Milling, CAD/CAM
Pekkton^®^ ivory, blank	PEKK	Cendres and Métaux, Biel/Bienne, Switzerland	Polyetherketoneketone (PEKK),	10% TiO_2_	5.1	200	Milling, CAD/CAM
Ultaire^®^ AKP	AKP	Myerson Tooth, Chicago, IL, USA	Arylketonepolymer (AKP)	No filler	3.5	148	Milling, CAD/CAM
breCAM.BioHPP blank	PEEK	Bredent, GmbH & Co KG, Senden, Germany	Polyetheretherketone (PEEK)	20% TiO_2_	>4.2	≥160	Milling, CAD/CAM
PEAK-Experiment No. 37	IM-E		PEEK, 5% TiO_2_ + 25% mixed	5% TiO_2_ + 25% mixed	-	-	Injection molded
breCAM.BioHPP (thp) ds2, Ch.: 488555	IM		PEEK, 12% TiO_2_ + 10% mixed	12% TiO_2_ + 10% mixed	-	-	Injection molded
BioHPP ds2, Ch.: 410368	P		PEEK, 15% TiO_2_ + 7% mixed	15% TiO_2_ + 7% mixed	-	-	Pressed
BioHPP plus ds2 Ch.: 410368	P plus		PEEK, 12% TiO_2_ + 11% mixed	12% TiO_2_ + 11% mixed	-	-	Pressed
Material: Vestakeep i4 3DF Printed by: Apium P220	A	Evonik Industries AG, Essen, Germany Apium Additive Technologies GmbH, Karlsruhe, Germany	PEEK	no filler	3.5	94	3D-FFF
Formlabs Permanent Crown, vertical	MA-V	Formlabs, Somerville, MA, USA Direction: 90° to building platform, Layer: 50 µm Cleaning: 3 min Isopropanol (99%) (Form Wash, Formlabs, USA) Polymerization: 2 × 20 min, 60 °C (Form Cure, Formlabs, USA)	Methacrylate (MA)	30–50%, diameter 0.7 µm	4.09	116	3D-SLA
Formlabs Permanent Crown, horizontal	MA-H			30–50%, diameter 0.7 µm	4.09	116	3D-SLA

**Table 2 materials-16-02373-t002:** Relative fluorescence intensities (mean and standard deviation; --: not determined) with bacteria (*S. mutans* and *S. sanguinis*) on different materials with different surface treatments (P0, P1000, and P4000).

		Relative Fluorescence Intensities					
Bacteria	Polishing	P0		P1000		P4000	
	Material	Mean	SD	Mean	SD	Mean	SD
*S. mutans*	TC	40.1	20.6	33.8	17.2	36.0	17.1
	BC	143.0	58.7	133.0	52.8	60.1	16.4
	AKP	235.8	190.0	154.6	115.4	123.9	43.1
	PEKK	18.5	80.4	64.6	50.9	44.2	38.0
	PEEK	62.3	55.3	86.0	43.3	162.3	34.5
	IM-E	212.8	111.7	247.0	188.9	257.3	157.3
	IM	120.9	132.7	77.2	28.7	127.4	52.6
	P	531.1	359.5	87.9	27.2	111.0	39.9
	P plus	495.3	340.7	120.3	88.6	121.8	52.2
	A	214.6	173.5	358.0	291.1	263.9	218.8
	MA-H	1844.1	588.2	189.8	78.2	318.7	107.3
	MA-V	5187.0	892.3	452.2	166.3	490.9	195.7
	Sinfony	--	--	--	--	236.6	216.9
	glass	143.4	112.3	--	--	--	--
*S. sanguinis*	TC	200.9	66.6	147.1	61.0	208.1	136.9
	BC	591.3	147.5	317.8	53.7	295.1	120.9
	AKP	14.8	122.6	101.6	56.1	133.3	67.8
	PEKK	281.1	120.3	254.9	97.1	228.5	78.2
	PEEK	156.8	86.9	249.1	80.0	266.6	87.4
	IM-E	138.7	70.3	125.9	34.7	148.9	31.0
	IM	124.3	59.2	131.6	92.9	161.8	74.7
	P	1268.9	737.7	493.2	512.5	343.7	347.6
	P plus	897.6	374.3	461.0	187.8	408.6	134.9
	A	161.3	54.7	299.9	193.8	301.6	136.0
	MA-H	1605.3	790.4	229.0	62.3	278.4	67.5
	MA-V	2949.1	1078.1	448.7	196.9	390.5	90.5
	Sinfony	--	--	--	--	216.5	228.7
	Glass	208.2	135.2	--	--	--	--

**Table 3 materials-16-02373-t003:** Surface parameters (R_z_, R_a_, and SFE) for different polishing protocols (mean and standard deviation) on different materials with different polishing (P0, P1000, and P4000).

	P0					P1000					P4000				
	R_z_ (µm)	R_z_ SD	R_a_ (µm)	Ra SD	SFE (mJ/m^2^)	R_z_ (µm)	Rz SD	R_a_ (µm)	Ra SD	SFE (mJ/m^2^)	R_z_ (µm)	Rz SD	R_a_ (µm)	Ra SD	SFE (mJ/m^2^)
TC	70.9	7.4	6.8	0.7	38.2	69.7	5.5	6.6	1.5	43.37	30.8	5.2	3.3	0.4	49.47
BC	68.3	5.8	4.8	0.7	44.83	52.1	4.4	3.8	0.3	36.84	36.2	6.0	3.4	1.0	32.79
PEKK	134	7.9	13.8	0.5	32.9	61.1	8.4	4.5	0.3	46.12	51.1	4.5	5.5	0.5	46.76
AKP	82	8.1	5.6	0.5	44.72	62.2	5.2	6.8	1.1	43.56	50.1	5.4	5.7	1.2	42.33
PEEK	98.1	3.5	9.4	1.1	32.18	84	7.7	9.0	1.9	36.94	48.2	5.4	5.2	0.8	43.27
IM-E	81.3	9.8	5.7	1.1	39.95	61.8	5.9	4.7	0.6	41.42	47.0	5.2	5.8	0.6	45.89
IM	64.8	3.7	5.4	0.8	48.57	62.4	4.3	5.6	0.8	43.11	38.2	2.9	4.1	0.2	46.64
P	86	7.4	6.4	0.8	44.1	65.7	4.6	6.0	0.9	35.68	27.7	3.3	3.0	0.4	45.34
P plus	98.5	7.4	7.6	0.4	32.18	62.2	4.8	5.0	0.5	36.94	38.7	4.5	4.4	0.5	36.66
A	59.7	12.2	3.6	0.9	58.55	69.6	4.8	7.9	0.1	48.36	58.6	8.0	6.7	0.9	44.56
MA-V	67	12.0	6.1	2.1	45.12	73.3	11.5	7.1	3.1	44.57	32.1	7.0	4.0	0.9	45.28
MA-H	78.8	14.1	7.3	2.9	42.31	64.4	5.0	4.6	0.5	40.97	36.8	3.2	3.9	0.3	45.72

## Data Availability

The data presented in this study are available upon request from the corresponding authors.

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
