# Peer review of "Bacterial Adhesion on Dental Polymers as a Function of Manufacturing Techniques"

_materials, 2023, doi:10.3390/ma16062373_

Round 1
Reviewer 1 Report
The paper focuses on the adhesion of Streptococcus mutans, Streptococcus sanguinis on dental materials made from polyaryletherketone, resin-based CAD/CAM and a 3D-printed methacrylate based materials. The bacterial adhesion (determined by means of a fluorometric assay ) is considered with respect to Surface roughness (determined by 3-D laser scanning microscope)
The paper falls within the scope of the journal. My comments are shown below:
The abstract currently contains elements that should be better included in Conclusions
The authors should define a number of abbreviations they use in the abstract and the main body of the paper.
Scale bars should be used in SEM images.
Author Response
We would like to thank you and the reviewers for reading and commenting on our paper. We have thoroughly addressed the issues raised by the reviewers and have followed their recommendations.
Changes are highlighted in the revised manuscript. In the text below, we outline and comment each changes made as raised in the reviewers´ comments.
We hope that the revisions are correct and in your interest!
The paper focuses on the adhesion of Streptococcus mutans, Streptococcus sanguinis on dental materials made from polyaryletherketone, resin-based CAD/CAM and a 3D-printed methacrylate based materials. The bacterial adhesion (determined by means of a fluorometric assay ) is considered with respect to Surface roughness (determined by 3-D laser scanning microscope)
The paper falls within the scope of the journal. My comments are shown below:
Comments: Thank you.
The abstract currently contains elements that should be better included in Conclusions
Comments: The abstract was rewritten (also according to the other reviewers) and individual elements were also used in the Conclusions section.
The authors should define a number of abbreviations they use in the abstract and the main body of the paper.
Comments: A number of abbreviations were defined and used in the abstract and the main body of the paper. (Streptococcus mutans (S. mutans) and Streptococcus sanguinis (S. sanguinis), Pearson correlation (PC), polyetheretherketone (PEEK, polyetherketoneketone (PEKK), arylketonepolymer (AKP), methacrylate (MA), additive (AM) and subtractive manufacturing (SM).
Scale bars should be used in SEM images.
Comments: scale bars are at the end of the SEM, but very small due to magnification. We added a red reference bar in the first figure, which represents 80 µm for every figure.
Reviewer 2 Report
Given the new form of the manuscript I agree that most issues have been resolved, including the graphic abstract.
Some minor issues remain concerning English and the Conclusion paragraph which is rather short.
Otherwise, the paper has acceptable scientific soundness, even if the innovation part/interest to readers is moderate.
Author Response
We would like to thank the reviewers for reading and commenting on our paper. We have thoroughly addressed the issues raised by the reviewers and have followed their recommendations.
Changes are highlighted in the revised manuscript. In the text below, we outline and comment each changes made as raised in the reviewers´ comments.
We hope that the revisions are correct and in your interest!
Given the new form of the manuscript I agree that most issues have been resolved, including the graphic abstract.
Some minor issues remain concerning English and the Conclusion paragraph which is rather short.
Otherwise, the paper has acceptable scientific soundness, even if the innovation part/interest to readers is moderate.
Comments: Thank you. The whole text was revised. Conclusion paragraph was improved. Some revision in English were made. “Resin-based materials, composites and PAEK showed noticeable differences in bacterial adhesion. Fabrication methods were shown to play a critical role for bacterial adhesion: the pressed PEEK show exhibited the highest initial accumulations, whereas . the milled PAEK samples provided similar lower adhesion. Horizontal DLP fabrication reduces bacterial adhesion in comparison to vertical processing. For DLP printed MA as well as for pressed PEEK, the surface finishing significantly reduces bacterial adhesion and is therefore essential for clinical application. The processing method as well as the polish affect the quality of the surfaces: Apium and Ultraire already have very smooth surfaces due to the processing and thus low bacterial adhesion. Horizontal DLP fabrication reduc-es bacterial adhesion. Roughness values <10 µm or polishing appear to be essential for reducing bacterial adhesion while the SFE did not show any influence.”
“Resin-based materials, composites and PAEK showed noticeable differences in bacterial adhesion. Fabrication methods were shown to play a critical role for bacterial adhesion: the pressed PEEK show exhibited the highest initial accumulations, whereas . the milled PAEK samples provided similar lower adhesion. Horizontal DLP fabrication reduces bacterial adhesion in comparison to vertical processing. For DLP printed MA as well as for pressed PEEK, the surface finishing significantly reduces bacterial adhesion and is therefore essential for clinical application. The processing method as well as the polish affect the quality of the surfaces: Apium and Ultraire already have very smooth surfaces due to the processing and thus low bacterial adhesion. Horizontal DLP fabrication reduc-es bacterial adhesion. Roughness values <10 µm or polishing appear to be essential for reducing bacterial adhesion while the SFE did not show any influence.”
Reviewer 3 Report
This is a well presented manuscript and the authors produced a significant amount of work in the number of materials tested and the different conditions.
Overall, is a topical article covering the full range of 3D printed and CAD/CAM materials in dentistry.
I have a few minor points:
Methodology should be better described. The authors mention 3D surface roughness but they only present 2D parameters. Also, there is no information about cut-off length, resolution, scanning area etc. The authors chose two roughness parameters (Ra and Rz), that are effectively the same (one is derived from the other). They should have both spacing and amplitude parameters for better characterisation.
For SEM there is no mention on how the specimens were prepared, imaging conditions, number of specimens etc.
For 3D printing, additional information on curing time/type, washing of specimens etc.
In the results, in some cases the roughness is increasing although specimens are more polished. Can you explain why?
Since the authors looked at two different printing orientations, the recent paper by Altarazi et al., should be cited [Dent Mater. 2022 Dec;38(12):1841-1854. ]
Author Response
We would like to thank you and the reviewers for reading and commenting on our paper. We have thoroughly addressed the issues raised by the reviewers and have followed their recommendations.
Changes are highlighted in the revised manuscript. In the text below, we outline and comment each changes made as raised in the reviewers´ comments.
We hope that the revisions are correct and in your interest!
This is a well presented manuscript and the authors produced a significant amount of work in the number of materials tested and the different conditions.
Overall, is a topical article covering the full range of 3D printed and CAD/CAM materials in dentistry.
Comments: Thank you.
I have a few minor points:
Methodology should be better described. The authors mention 3D surface roughness but they only present 2D parameters. Also, there is no information about cut-off length, resolution, scanning area etc. The authors chose two roughness parameters (Ra and Rz), that are effectively the same (one is derived from the other). They should have both spacing and amplitude parameters for better characterisation.
Comments: The methodology was improved and corrected. Additional information was provided. Standard amplitude parameters characterize the surface based on the vertical deviations of the roughness profile from the mean line. Spacing parameters give how often the profile crosses certain thresholds. A conversion from Rz to Ra is only possible in a conversion interval that contains all possible values. Ra and Rz are classically used to characterize surfaces before bacterial colonization. These values can therefore be compared with the literature, which seems important to compare results with earlier investigations. As the present results show, the two parameters have different relevance for the results. The differentiation therefore appears to make sense. Both values relate in part to bacterial adhesion. Therefore, we have displayed both values. Distance parameters describe how often the profile exceeds certain thresholds, but are not used in the literature in relation to microbial growth, so we did not use these values. Hope the reviewer can accept our attitude.
Roughness (Rz, Ra) on all specimens was measured with a laser scanning microscope (KJ VK-X100K, Keyence, J, class 2 laser 658 nm, 50× magnification; measuring range 2500x1900 µm, cut-off λs 0.8µm λc 0.08 µm, resolution h= 0,005 µm, w= 0,01 µm, 50 x, 12 nm, repeatability σ h: 0,02 µm w: 0,05 µm).
For SEM there is no mention on how the specimens were prepared, imaging conditions, number of specimens etc.
Comments: The required additional information was provided in M&M. SEM micrographs (Phenom, FEI Company, NL) were used to assess the surfaces prior and after polishing (untreated specimens, resolution 1024, quality high, WD 10µm, magnifications up to 1500x, n=2).
For 3D printing, additional information on curing time/type, washing of specimens etc.
Comments: The required additional information was provided in table 1. Formlabs, Somerville, USA Direction: 90° to building plat-form, Layer: 50 µm, Cleaning: 3min Isopropanol (99%), (Form Wash, Formlabs, USA), Polymerization:2x 20 min, 60°C, (Form Cure, Formlabs, USA)
In the results, in some cases the roughness is increasing although specimens are more polished. Can you explain why?
These observations can be explained by the distinct type of fabrication, as the materials showed smooth surfaces directly after production. For material A, this is certainly due to the filament printing process, where the processing parameters such as temperature and the type of die through which the molten material is pressed, determine the surface quali-ty. For the AKP specimens, it is expected that the different polymer matrix compared to the other PAEK, and for IM-E the relatively high filler content for PEEK influence the surface finish.
We added this information in the text.
Since the authors looked at two different printing orientations, the recent paper by Altarazi et al., should be cited [Dent Mater. 2022 Dec;38(12):1841-1854. ]
We added this citation in the text. Altarazi A, Haider J, Alhotan A, Silikas N, Devlin H. Assessing the physical and mechanical properties of 3D printed acrylic material for denture base application. Dent Mater. 2022 Dec;38(12):1841-1854. doi: 10.1016/j.dental.2022.09.006. Epub 2022 Oct 1. PMID: 36195470.